# BIChromET: A Chromogenic Culture Medium for Detection of Piperacillin/Tazobactam and Cefepime Resistance in *Pseudomonas aeruginosa*

**DOI:** 10.3390/antibiotics12111573

**Published:** 2023-10-28

**Authors:** José Manuel Ortiz de la Rosa, Ángel Rodríguez-Villodres, Guillermo Martín-Gutiérrez, Carmen Cintora Mairal, José Luis García Escobar, Lydia Gálvez-Benítez, José Miguel Cisneros, José Antonio Lepe

**Affiliations:** 1Clinical Unit of Infectious Diseases, Microbiology and Parasitology, University Hospital Virgen del Rocío, 41013 Seville, Spain; jmanuel.ortiz@juntadeandalucia.es (J.M.O.d.l.R.); guillermo.martin.gutierrez.sspa@juntadeandalucia.es (G.M.-G.); carmen.cintora.sspa@juntadeandalucia.es (C.C.M.); josel.garcia.escobar.sspa@juntadeandalucia.es (J.L.G.E.); lidia.galvez.sspa@juntadeandalucia.es (L.G.-B.); josea.lepe.sspa@juntadeandalucia.es (J.A.L.); 2Institute of Biomedicine of Seville (IBiS), University Hospital Virgen del Rocío/CSIC/University of Seville, 41013 Seville, Spain; 3Centro de Investigación Biomédica en Red de Enfermedades Infecciosas (CIBERINFEC), Instituto de Salud Carlos III, Madrid 28029, Spain; 4Department of Health Sciences, Loyola Andalucía University, 41704 Seville, Spain; 5Department of Medicine, Faculty of Medicine, University of Seville, 41009 Seville, Spain; 6Department of Microbiology, Faculty of Medicine, University of Seville, 41009 Seville, Spain

## Abstract

Objectives: The BIChromET selective medium for detecting piperacillin-tazobactam (TZP) and cefepime (FEP) resistant *Pseudomonas aeruginosa* was developed. Methods: The performance of this medium was first evaluated using a collection of 100 *P. aeruginosa* clinical strains (70 TZP-susceptible, 30 TZP-resistant, 58 FEP-susceptible, and 42 FEP-resistant). Then, we performed clinical validation by testing 173 respiratory clinical samples. Results: The BIChromET medium showed excellent sensitivity (TZP (avg. 96.7%); FEP (avg. 92.7%)) and specificity (TZP (avg. 98.9%); FEP (avg. 98%)) in distinguishing the detection limit ranging from 10^4^ to 10^8^ CFU/mL. Then, testing the bronchoalveolar lavage (BAL) and tracheobronchial aspirate (TBA) clinical specimens (N = 173) revealed the excellent performance of the medium with *P. aeruginosa*, showing 100% and 92.6% of categorical agreements with the results obtained via the broth microdilution methods (BMD) for TZP and FEP, respectively. Conclusion: This medium allows for easy and accurate detection of TZP/FEP-resistant isolates regardless of their resistance mechanisms.

**Keywords;** piperacillin/tazobactam; cefepime; empiric treatment; *Pseudomonas aeruginosa*

## 1. Introduction

*Pseudomonas aeruginosa* is one of the most common causes of pneumonia in hospitalized and immunocompromised patients, frequently associated with high morbidity and mortality [1,2,3]. According to the National Healthcare Safety Network (NHSN) and the EU-VAP/CAP study, *P. aeruginosa* is the second most prevalent microorganism isolated from nosocomial pneumonia [4,5]. *P. aeruginosa* infections are a serious concern in hospitals. Patients in critical conditions can die from pneumonia caused by *P. aeruginosa*, and the elimination of *P. aeruginosa* is very difficult because of its wide variety of resistance mechanisms [6]. Furthermore, it is responsible for considerable additional healthcare costs and resource utilization due to the difficulty in dealing with *P. aeruginosa* infections [7].

The initial management of patients with *P. aeruginosa* infections, especially from the intensive care unit (ICU), involves obtaining culture results and administering appropriate antibiotics within the first hour [8]. Choosing an appropriate initial antibiotic is essential in managing *P. aeruginosa* infections, as the inappropriate initial choice of antimicrobial regimen has previously been shown to be associated with increased mortality [9,10,11,12]. A previous retrospective study found that initial inappropriate antimicrobial therapy in septic shock resulted in a 5-fold reduction in survival [13]. A large meta-analysis of patients with severe bacterial infections showed that patients who received appropriate initial antibiotic therapy had lower treatment failure rates, shorter hospital length of stay and cost, and lower mortality rates than patients who received inappropriate initial antibiotic therapy [14].

Two of the most used antibiotics for the empirical treatment of hospital-acquired *P. aeruginosa* infections are piperacillin-tazobactam (TZP) and cefepime (FEP) [15]. The initial empirical treatment decisions are challenged by the lack of knowledge of the bacterial resistance profile. The reference microbiological diagnostic tools available for bacteria causing respiratory tract infections may extend the time to obtain the results to 24–48 h [16] (Figure 1). During this period, patients may be receiving inappropriate treatment, increasing morbidity and mortality.

Currently, antimicrobial stewardship programs (ASPs) are being implemented. ASPs aim to promote the safe, effective, and efficient use of antibiotics to optimize patient care while mitigating individual patient risk and the impact of antimicrobial use on the greater population [17,18,19]. These kinds of programs encompass prevention, early treatment, and rapid diagnosis. Therefore, it is very important for the prognosis of the patients to have rapid diagnostic techniques to guide empirical treatment in the shortest possible time. Molecular techniques based on multiplex PCR have been developed in recent years to identify respiratory pathogens directly from the samples [20]. However, molecular techniques present a limitation due to the discrepancy between genotype and phenotype, especially in *P. aeruginosa*, which could lead to misinterpreting the results [21]. Thus, the main challenge in respiratory infections is rapidly detecting the antibiotic resistance profile. In this sense, we aimed to develop a selective culture medium for screening TZP/FEP-resistant *P. aeruginosa* regardless of the corresponding resistance mechanism in 24 h, reducing to half the time to optimize the empirical treatment (Figure 1).

## 2. Results

According to the minimal inhibitory concentration (MIC) results obtained via broth microdilution method (BMD), this collection included 70 TZP-susceptible isolates + 30 TZP-resistant isolates, and 59 FEP-susceptible isolates + 41 FEP-resistant isolates (Appendix A).

On the FEP part of the bi-plates, the results showed that none of the FEP-susceptible isolates grew on the BIChromET medium, except for five isolates of 1 × 10^8^ CFU/mL of bacterial concentrations. On the other hand, all the FEP-resistant isolates were recovered within 24 h on the BIChromET medium using an inoculum between 1 × 10^6^ and 1 × 10^8^ CFU/mL. However, 11 and 3 FEP-resistant isolates did not grow onto the BIChromET selective plates when inoculated at 1 × 10^4^ and 1 × 10^5^ CFU/mL of bacterial concentration, respectively (Table 1). For the TZP part of the bi-plates, the results showed that all the TZP-susceptible isolates did not grow onto the BIChromET medium, except for four isolates at 1 × 10^8^ CFU/mL. On the other hand, all the TZP-resistant isolates were recovered within 24 h on BIChromET medium using an inoculum between 1 × 10^5^ and 1 × 10^8^ CFU/mL. However, four TZP-resistant isolates did not grow onto the BIChromET selective plates when inoculated at 1 × 10^4^ CFU/mL (Table 1). These data allowed us to set the sensitivity and specificity of the medium in each dilution tested. The sensitivity values varied between the different dilutions but remained above 90% in all the bacterial concentrations tested, most being 100%, except at 1 × 10^4^ CFU/mL. On the other hand, the BIChromET plates showed 100% in the specificity of all bacterial concentrations evaluated, except at 1 × 10^8^ CFU/mL (TZP and FEP) and 1 × 10^7^ CFU/mL (only FEP), in which a decrease in specificity [FEP (10^7^: 98.3% and 10^8^: 91.54%) and TZP (10^8^: 94.3%)] was observed (Table 1).

Among the 173 clinical specimens (bronchoalveolar lavage (BAL)/tracheobronchial aspirate (TBA) analyzed, 27 *P. aeruginosa* were recovered via the conventional techniques used in the Microbiology Service at the University Hospital Virgen del Rocío (Seville). The BIChromET plates, compared to the reference BMD, showed a high level of categorical agreement (CA), reaching 100% and 92.6% for TZP and FEP, respectively. In particular, the 7.4% loss in categorical agreement observed in FEP corresponds to two major errors (MEs, defined as false resistance results) (Appendix A. Despite the reduced number of *P. aeruginosa* (N = 27) recovered in the validation step with 173 clinical specimens (BAL and TBA), the data obtained with the BIChromET plates and BMD allowed us to calculate the sensitivity and specificity of the media. TZP showed a sensitivity and specificity of 100% (IC_sensitivity_: 81.6%–100%; IC_specificity_: 72.2–100%), and FEP showed a sensitivity of 100% (IC: 78.5%–100%) and a specificity of 84.6% (IC: 57.8–95.7%) (Table 2).

Moreover, our media could detect some FEP/TZP-resistant Gram-negative bacteria that were also identified using conventional methods, such as MicroScan Walkaway (Beckman Coulter, Brea, Callifornia, USA) (Appendix A). Notably, no competing flora (which are frequently found on standard medium), such as Gram-positive bacteria or fungi, was identified with the BIChromET plates after 24 h of incubation, indicating that this medium has high levels of specificity and selectivity for TZP/FEP-resistant Gram-negative bacteria.

## 3. Discussion

In this study, we have developed the BIChromET selective medium, being the first medium that could detect TZP- and FEP-resistant *P. aeruginosa* isolates and reduce the time to adjust the antimicrobial therapy in 24 h in severe respiratory infections. Although the BIChromET medium showed excellent sensitivity and specificity during the evaluation, limitations were found when some isolates were inoculated at 1 × 10^4^ and 1 × 10^8^ CFU/mL, decreasing the sensitivity and specificity values, respectively (Table 1). The reduction in the sensitivity was due to the isolates with MIC values close to the breakpoint but still resistant (FEP (16 μg/mL) or TZP (32 μg/mL)) that did not grow at 1 × 10^4^ CFU/mL. In contrast, the slight reduction in the specificity was due to the isolates with MIC values close to the breakpoint but still susceptible (FEP (8 μg/mL) or TZP (16 μg/mL)) that grew at 1 × 10^8^ CFU/mL. A diagnostic algorithm was created to address the limitations presented above, especially the one significantly affecting the sensitivity, and to minimize the possible false negative obtained with the BIChromET medium (Figure 1B). This algorithm ignores plates with an absence of growth when the blood agar medium shows a bacterial concentration of 1 × 10^4^ CFU/mL. It is noteworthy that several studies and data from the University Hospital Virgen del Rocío showed that more than 95% of positive BAL and TBA contain a bacterial concentration ≥ 1 × 10^5^ CFU/mL [22,23]. This fact proves that the test will be useful in at least 95% of the samples received in the Microbiology service of the University Hospital Virgen del Rocío. Another limitation is the absence of MIC values since our media only determines susceptible or resistant bacteria. Therefore, our media will categorize as susceptible all isolates whose MIC values are ≤16 μg/mL. Notwithstanding, EUCAST guidelines recommended high doses of cefepime and piperacillin-tazobactam for all susceptible P. aeruginosa.

This medium constitutes a useful tool combo for identifying TZP/FEP-resistant strains and possibly prevents further dissemination and outbreaks by screening patients potentially infected with such resistant strains. Unlike other tests, the medium allows direct screening from clinical samples, such as BAL and TBA, usually collected from severe infection patients. The selectivity of the BIChromET medium for TZP/FEP-resistant bacteria is not impacted by the mixed flora found in many clinical samples. Moreover, the BIChromET plates showed excellent performance with respiratory samples (BAL and TBA). Using the BIChromET plates as a complement to the traditional techniques may play an important role in clinical decision-making, optimizing antibiotic usage. In 2022, *P. aeruginosa* encompassed 18.6% of all respiratory isolates (N = 4925) in our institution. Noteworthy, in 2022, 27% and 23% of the *P. aeruginosa* isolates were resistant to FEP and TZP, respectively. In this way, in around 25% of the patients with respiratory infections by *P. aeruginosa*, the BIChromet medium will allow to optimize the empirical treatment 24 h earlier than with the conventional methods. On the other hand, in 75% of the patients, our media will allow maintaining or simplifying the empirical treatment of the patients, preserving the utility of the last-resort antibiotics’ usefulness. Thus, this selective medium could cover a need of many patients since TZP and FEP have become two of the first options against hospital-acquired respiratory infections caused by *P. aeruginosa* bacteria in many places and the lack of rapid diagnostic tests in such infections [16].

In conclusion, the BIChromET medium showed excellent performance and was easy to prepare. Its cost of production would be around EUR 0.70/plate, maintaining a great cost–benefit balance. Moreover, the BIChromET medium would be added to the diagnostic toolbox dedicated to the rapid detection of antibiotic resistance in respiratory infections, which may help to optimize the empirical treatment, where the delay of the appropriate therapy is crucial for the outcome of the patients. Notwithstanding, further clinical evaluations of this medium will now be needed in daily clinical practice to further assess its usefulness.

## 4. Materials and Methods

**Bacterial strains.** A total of 100 *P. aeruginosa* isolates collected from different clinical samples (bronchoalveolar lavage (BAL), tracheobronchial aspirate (TBA), cerebrospinal fluid (CSF), blood cultures, sputum, wounds, and urine) from the Microbiology Service at the University Hospital Virgen del Rocío (Seville, Spain) were used in developing the BIChromET plates. This collection included 30 TZP-resistant isolates, 70 TZP-susceptible isolates, 41 FEP-resistance isolates, and 59 FEP-susceptible isolates (Appendix A). Additionally, the BIChromET plates were clinically tested on 173 clinical specimens (BAL and TBA) recovered over 3 months. *P. aeruginosa* PAO1 was used as control negative, and 2 TZP/FEP-resistance clinical isolates were used as control positive. Additionally, *P. aeruginosa*, *Stenotrophomonas maltophilia*, *Acinetobacter baumannii*, *Klebsiella pneumoniae*, and *Escherichia coli* were used to establish the color change of the different species when growth on the BIChromET plates. The isolates were not screened at the molecular level for the presence of resistance determinants due to the medium being developed to detect resistant isolates independently of their resistance mechanism.

**Susceptibility testing.** MIC values for TZP and FEP were determined using the broth microdilution method following the EUCAST recommendations (https://www.eucast.org/ast_of_bacteria/mic_determination (accessed on 1 February 2023)). Clinical breakpoints were established according to the European Committee on Antimicrobial Susceptibility Testing (EUCAST; 2023) [24]. Hence, isolates with TZP MICs among >0.001 μg/mL and ≤16 μg/mL were categorized as I (susceptible, increased exposure), while those with MICs of >16 μg/mL were categorized as resistant, and isolates with FEP MICs among >0.001 μg/mL and ≤8 μg/mL were categorized as I (susceptible, increased exposure), while those with MICs of >8 μg/mL were categorized as resistant. EUCAST established the susceptible breakpoint at ≤0.001 μg/mL to categorize all susceptible *P. aeruginosa* as I. Thus, therapeutic success with susceptible *P. aeruginosa* would depend on the dosing regimen and/or the site of infection.

**Selective medium for TZP and FEP resistance.** CLED agar medium (reference; Biomerieux, Paris, France) was used for optimal screening following the manufacturer’s instructions. After testing several culture conditions, a selective medium was set up and supplemented with piperacillin, tazobactam, and FEP (Sigma-Aldrich, St. Louis, MO, USA) at 24, 4, and 4 µg/mL, respectively. Moreover, vancomycin (Duchefa Biochemie, Haarlem, Netherlands) and amphotericin B (Acros Organics, Morris Plains, New Jersey, USA) were added to the medium at a final concentration of 20 and 5 µg/mL to prevent the growth of Gram-positive bacteria and fungi, respectively. Additionally, a phenol red solution (0.5%) was added to the FEP section of the bi-plate to differentiate both sections visually. Thus, in the TZP section of the bi-plate (standard CLED medium), the lactose fermenter bacteria produced light beige colonies, unlike non-fermenters, which yield blue colonies. On the other hand, in the FEP part of the BIChromET medium (CLED with phenol red), the lactose fermenter bacteria produced yellow colonies, unlike non-fermenters, which yield purple colonies (Appendix A). The CLED powder was diluted in distilled water and autoclaved at 121 °C for 15 min. The antibiotic stock solutions were added when the medium reached 56 °C (Table 3). The prepared plates of this BIChromET medium were stored at 4 °C and were protected from direct light exposure before use for as long as 2 weeks. *Candida albicans*, *Staphylococcus aureus*, and the TZP/FEP-susceptible *E. coli* ATCC 25922 reference strain were sub-cultured daily on the BIChromET selective plates from a single batch of plates kept at 4 °C to test the shelf life of the medium. For at least 14 days, no growth could be observed.

**Evaluation assay.** The sensitivity and specificity cut-off values for detecting TZP- and FEP-resistant *P. aeruginosa* were established at 1 × 10^3^ CFU/mL, considering the positive results of only the sample, of which the isolates were recovered onto the BIChromET selective medium, which were plated at concentrations corresponding to >1 × 10^3^ CFU/mL. This cut-off value was fixed considering that BAL and TBA are positive when the ≥1 × 10^4^ CFU/mL bacteria are recovered from the clinical sample. Starting with a 0.5 McFarland standard (an inoculum of 1.5 × 10^8^ CFU/mL), serial 10-fold dilutions were made in 0.85% saline solution, and 100 µL aliquots of each dilution from 10^4^ to 10^8^ CFU/mL were plated onto the BIChromET selective medium. To quantify the viable bacteria in each dilution step, tryptic soy agar plates were inoculated concomitantly with 100 µL of each suspension and incubated overnight at 37 °C. Viable colonies were counted the following day. When no growth was observed after 18 h, incubation was extended up to 48 h to assess the negativity of the culture. The medium was designed to detect resistant *P. aeruginosa* from 10^4^ to 10^8^ CFU/mL based on the IDSA guidelines [3]. It established that a BAL culture was positive when more than 10^4^ CFU/mL bacteria were recovered from the BAL. After evaluation with a collection of 100 *P. aeruginosa*, the results were analyzed for each dilution of bacterial concentration used in this study.

**Validation experiments.** To check the performance of the selective medium in clinical samples, a total of 173 samples obtained from patients admitted to the University Hospital Virgen del Rocío (21 BAL and 152 TBA clinical specimens) were tested using the BIChromET plates. Then, 100 μL of the BAL or TBA were inoculated onto each half of the BIChromET plates and incubated overnight at 37 °C. The colonies of different morphologies, sizes, and colors from each plate were selected for further experiments, such as identification (mass spectrometry) and resistance phenotype (gradient strips (Liofilchem, Italy) for non-*P. aeruginosa* isolates) (Appendix A). To confirm the resistance patterns in *P. aeruginosa*, the grown colonies of *P. aeruginosa* were tested for TZP and FEP susceptibility using BMD. The results were interpreted according to the EUCAST 2023 breakpoints [24].

**Statistical analysis.** The specificity (proportion of TZP- or FEP-susceptible isolates that are correctly determined) and the sensitivity (proportion of TZP- or FEP-resistant isolates that are correctly determined) were calculated for each dilution used in the evaluation step to know the limitation of the medium, its performance at the different concentration, and how to overcome these limitations using broth microdilution as the gold standard method. On the other hand, 95% confidence intervals (CI), a predictive positive value (PPV), and a predictive negative value (PNV) were also estimated. For the validation step, the sensitivity, specificity, and 95% CIs were calculated with the clinical specimens (BAL and TBA) to detect the resistance in *P. aeruginosa* using broth microdilution as the gold standard method. In the clinical evaluation step, true positive results were considered when a TZP or FEP-resistant *P. aeruginosa* (BMD MIC) growth in the BIChromET plates, while the absence of growth in the BIChromET plates when a TZP or FEP-susceptible *P. aeruginosa* (MD MIC) were present in the BAL/TBA samples were considered true negative. Moreover, the endpoints were considered in categorical agreement when the results were in the same susceptibility category (regardless of the MIC) for *P. aeruginosa*. VME is a very major error (false susceptibility), and ME is a major error (false resistance).

## Figures and Tables

**Figure 1 antibiotics-12-01573-f001:**
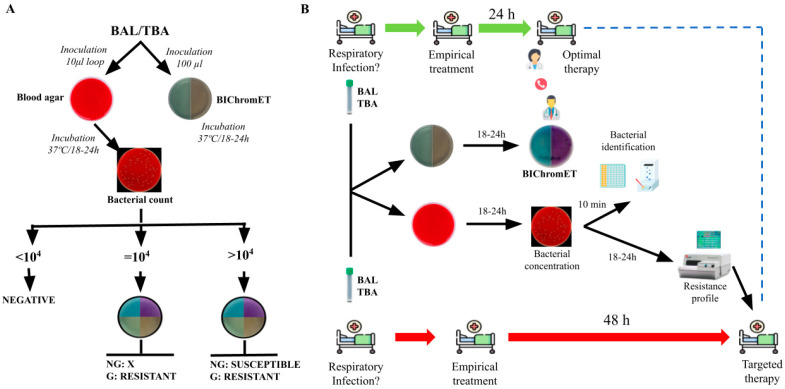
(**A**) Diagnostic algorithm designed for using the BIChromET plates in clinical practice. (**B**) Flowchart of the BIChromET medium methodology and its possible application in clinical practice. BAL, Bronchoalveolar lavage; TBA, Tracheobronchial aspirate; G, growth; NG, not growth.

**Table 1 antibiotics-12-01573-t001:** Performance of the BIChromET medium for detecting *P. aeruginosa* resistance or susceptibility to TZP and/or FEP from bacterial colonies.

		Bacterial Concentrations (CFU/mL)				
	Gold Standard	1 × 10^4^	1 × 10^5^	1 × 10^6^	1 × 10^7^	1 × 10^8^
Cefepime						
Susceptible	59	70	63	59	58	54
Resistance	41	30	37	41	42	46
Sensitivity (%)	-	73.2	90.2	100	100	100
CI (%)	-	58.1–84.3	77.5–96.1	91.4–100	91.4–100	91.4–100
Specificity (%)	-	100	100	100	98.3	91.5
CI (%)	-	93.9–100	93.9–100	93.9–100	91–99.7	81.6–96.3
PPV (%)	-	100	100	100	97.6	89.1
NPV (%)	-	84.3	93.7	100	100	100
Piperacillin-Tazobactam						
Susceptible	70	74	71	70	70	66
Resistance	30	26	29	30	30	34
Sensitivity (%)	-	86.7	96.7	100	100	100
CI (%)	-	70.3–94.7	83.3–99.4	88.6–100	88.6–100	83.8–99.4
Specificity (%)	-	100	100	100	100	94.3
CI (%)	-	94.8–100	94.8–100	94.8–100	94.8–100	86.2–97.8
PPV (%)	-	100	100	100	100	88.2
NPV (%)	-	94.6	98.6	100	100	100
		Bacterial Concentrations (CFU/mL)				
	Gold Standard	1 × 10^4^	1 × 10^5^	1 × 10^6^	1 × 10^7^	1 × 10^8^
Cefepime						
Susceptible	59	70	63	59	58	54
Resistance	41	30	37	41	42	46
Sensitivity (%)	-	73.2	90.2	100	100	100
CI (%)	-	58.1–84.3	77.5–96.1	91.4–100	91.4–100	91.4–100
Specificity (%)	-	100	100	100	98.3	91.5
CI (%)	-	93.9–100	93.9–100	93.9–100	91–99.7	81.6–96.3
PPV (%)	-	100	100	100	97.6	89.1
NPV (%)	-	84.3	93.7	100	100	100
Piperacillin-Tazobactam						
Susceptible	70	74	71	70	70	66
Resistance	30	26	29	30	30	34
Sensitivity (%)	-	86.7	96.7	100	100	100
CI (%)	-	70.3–94.7	83.3–99.4	88.6–100	88.6–100	83.8–99.4
Specificity (%)	-	100	100	100	100	94.3
CI (%)	-	94.8–100	94.8–100	94.8–100	94.8–100	86.2–97.8
PPV (%)	-	100	100	100	100	88.2
NPV (%)	-	94.6	98.6	100	100	100

TZP, piperacillin/tazobactam, FEP, cefepime; CI, 95% confidence Interval, PPV, positive predictive value, NPV, negative predictive value.

**Table 2 antibiotics-12-01573-t002:** Validation of the BIChromET medium for detecting *P. aeruginosa* resistance or susceptibility to TZP and/or FEP from clinical samples.

	BIChromET	BMD
Cefepime:		
Susceptible	11	13
Resistance	16	14
CA	92.6%	
Errors	MEs (N = 2)	
Sensitivity ^(a) CI^	100% (78.5–100%)	
Specificity ^(a) CI^	84.6% (57.8–100%)	
Piperacillin-Tazobactam:		
Susceptible	10	10
Resistance	17	17
CA	100%	
Errors	-	
Sensitivity ^(a) CI^	100% (81.6–100%)	
Specificity ^(a) CI^	100% (72.2–100%)	

TZP, piperacillin/tazobactam, FEP, cefepime; CI, 95% confidence Interval; CA, Categorical agreement; BMD, broth microdilution. ^(a)^ Sensitivity and specificity were exclusively calculated using the gold standard method (BMD) as a reference.

**Table 3 antibiotics-12-01573-t003:** Preparation of the BICromET medium.

Compound	Stock Solution	Quantity or Vol Added	Final Concentration
Piperacillin-Tazobactam			
CLED agar medium		14.46 g	
Distilled water		400 mL	
Piperacillin	50 mg/mL in water	0.192 mL	24 mg/L
Tazobactam	50 mg/mL in water	0.032 mL	4 mg/L
Vancomycin	50 mg/mL in water	0.16 mL	20 mg/L
Amphotericin B	10 mg/mL in DMSO	0.2 mL	5 mg/L
Cefepime			
CLED agar medium		14,46 g	
Distilled water		400 mL	
Phenol red	0.5%	1 mL	0.00125%
Cefepime	50 mg/mL in water	0.032 mL	4 mg/L
Vancomycin	50 mg/mL in water	0.16 mL	20 mg/L
Amphotericin B	10 mg/mL in DMSO	0.2 mL	5 mg/L

DMSO, Dimehyl sulfoxide, TZP, piperacillin/tazobactam, FEP, cefepime; CI, 95% confidence Interval, PPV, positive predictive value, NPV, negative predictive value.

## Data Availability

The datasets used and/or analyzed during the present study are available from the corresponding author upon reasonable request.

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
