# Peer review of "BIChromET: A Chromogenic Culture Medium for Detection of Piperacillin/Tazobactam and Cefepime Resistance in Pseudomonas aeruginosa"

_antibiotics, 2023, doi:10.3390/antibiotics12111573_

Round 1

Reviewer 1 Report

Comments and Suggestions for Authors

The publication describes the developement of new medium BIChromET that is selective for the detection of piperacillin-tazobactam (TZP) and cefepime (FEP) resistant Pseudomonas aeruginosa in clinical specimens: bronchoalveolar lavage and tracheobronchial aspirate of patients with symptoms of pneumonia. The main advantage of this medium is shortened diagnosis time of patients with pneumonia necessary to initiate antibiotic therapy. The other benefit is the detection of resistance regardless of the mechanism of resistance.

The disadvantage of the method is the evident dependence of the sensitivity and specificity on the inoculum size.

More simplified and unequivocal protocol should be prepared for laboratory practice.

Incorrect table numbering in the text.

No line numbering makes the correction difficult.

Explanation of abbreviations should be located directly in Table 1 and 2 and also in supplementary materials.

Was the effect  of vancomycin  and amphotericin on Pseudomonas growth tested ?

The calculations in tables in the text are difficult to explain without supplementary data.

Is that possible to make the tables more concise and clear.

It is clear that before introduction of the BIChromET medium  to  clinical practice is necessary to test more patients both with hospital- and community-acquired pneumonia.

Author Response

The publication describes the developement of new medium BIChromET that is selective for the detection of piperacillin-tazobactam (TZP) and cefepime (FEP) resistant Pseudomonas aeruginosa in clinical specimens: bronchoalveolar lavage and tracheobronchial aspirate of patients with symptoms of pneumonia. The main advantage of this medium is shortened diagnosis time of patients with pneumonia necessary to initiate antibiotic therapy. The other benefit is the detection of resistance regardless of the mechanism of resistance.

The disadvantage of the method is the evident dependence of the sensitivity and specificity on the inoculum size.

R: We agree that the reduction of the sensitivity and specificity mainly when the inoculum is 10^4 is a limitation of BIChromET plates, due to this, we elaborated a diagnostic algorithm in order to minimize the false negative result. Everything was clarified in the manuscript (discussion) as follows: “Regarding the limitations presented above, especially the one significantly affecting the sensitivity and in order to minimize the possible false negative obtained with the BIChromET medium, a diagnostic algorithm was created ignoring plates with the absence of growth when the blood agar medium shows a bacterial concentration of 1 x 104 CFU/ml (Figure 1B).”

More simplified and unequivocal protocol should be prepared for laboratory practice.

R: The protocol fixed for the clinical laboratory excludes the preparation of the media. However, to clarify the protocol, some specifications have been made in the manuscript. Thus, the workers just have to inoculate 100 μl of the clinical sample onto each half of BIChromET plates and incubate it overnight at 37°C. Then, 24 hours later, we could observe the presence or absence of growth in the BIChromET medium (see manuscript).

Incorrect table numbering in the text.

R: This was an error in the process of adaptation to the journal format. Sorry for the inconvenience. The mistake was amended in the manuscript.

No line numbering makes the correction difficult.

R: The original manuscript contains the line numbering but after the transition to journal format, the line numbering disappears.

Explanation of abbreviations should be located directly in Table 1 and 2 and also in supplementary materials.

R: This section was amended on the Tables and Supplementary material.

Was the effect  of vancomycin  and amphotericin on Pseudomonas growth tested ?

R: Yes, the growth of Pseudomonas was tested with both antibiotics, individually and in combination, and no effect was observed.

The calculations in tables in the text are difficult to explain without supplementary data.

R: In relation to the supplementary data, three tables encompassing the raw data of the evaluation and validation of the plates were included in the submission of the manuscript. It should have been available for the reviewer. If necessary you could ask the editor about those supplementary materials.

Is that possible to make the tables more concise and clear.

R: As suggested by the reviewer, the tables were revised and simplified. Additionally, the abbreviations were added at the end of the tables. 

It is clear that before introduction of the BIChromET medium  to  clinical practice is necessary to test more patients both with hospital- and community-acquired pneumonia.

R: We agree with this affirmation. For this reason, our last sentence in the discussion highlighted that further clinical evaluations are needed in clinical practice to further assess its usefulness.

Reviewer 2 Report

Comments and Suggestions for Authors

De la Rosa et al report on their development of a chromogenic medium (BIChromET) that detects resistance of Pseudomonas aeruginosa to three antibiotics relevant in treatment of P. aeruginosa infections – piperacillin, tazobactam, and cefepime. Besides the study being innovative, the study is expected to have a high clinical impact. The authors provided enough background and were coherent in their presentation. Especially important is the clarity in the statement of their rationale, and their highlighting of the limitation of molecular techniques in the rapid detection of respiratory pathogens. The references cited are relevant, the results are clearly presented, and the conclusion are based on the results. The authors could, however, consider moving the following from the Results to the Methods section, as that is where they fit:

The first and last sentences of Paragraph 1

The first three sentences of Paragraph 2

The first, second, and fourth sentences of Paragraph 3.

The last paragraph

Comments on the Quality of English Language

Moderate edits are needed.

Author Response

De la Rosa et al report on their development of a chromogenic medium (BIChromET) that detects resistance of Pseudomonas aeruginosa to three antibiotics relevant in treatment of P. aeruginosa infections – piperacillin, tazobactam, and cefepime. Besides the study being innovative, the study is expected to have a high clinical impact. The authors provided enough background and were coherent in their presentation. Especially important is the clarity in the statement of their rationale, and their highlighting of the limitation of molecular techniques in the rapid detection of respiratory pathogens. The references cited are relevant, the results are clearly presented, and the conclusion are based on the results. The authors could, however, consider moving the following from the Results to the Methods section, as that is where they fit:

The first and last sentences of Paragraph 1

The first three sentences of Paragraph 2

The first, second, and fourth sentences of Paragraph 3.

The last paragraph

R: Following the recommendation of the reviewer, some parts of the results were included in the material and methods.

Reviewer 3 Report

Comments and Suggestions for Authors

The authors highlight the potential of the BIChromET medium as a valuable tool for the accurate detection of TZP and FEP-resistant P. aeruginosa strains, irrespective of the mechanisms underlying their resistance. Such innovations in diagnostic microbiology are pivotal in the battle against antibiotic resistance and in ensuring appropriate treatment for patients. This research holds great promise for improving clinical practices and patient care. Overall, the manuscript is well-structured and informative. However, some modifications are needed to enhance the manuscript's quality as follows:

1.     The Author mentions on page 12, "dilution from 104 to 108 UFC/ml." Could you kindly clarify the precise meaning of "UFC" in this context?

2.     Please maintain a consistent format for "CFU/mL" or "CFU/ml" across the entire manuscript.

3.     It is recommended that the Author includes a section discussing limitations and a clear conclusion in the manuscript to enhance comprehension.

4.     It is advisable to strictly follow the referencing and citation style specified by the journal's guidelines.

Author Response

The authors highlight the potential of the BIChromET medium as a valuable tool for the accurate detection of TZP and FEP-resistant P. aeruginosa strains, irrespective of the mechanisms underlying their resistance. Such innovations in diagnostic microbiology are pivotal in the battle against antibiotic resistance and in ensuring appropriate treatment for patients. This research holds great promise for improving clinical practices and patient care. Overall, the manuscript is well-structured and informative. However, some modifications are needed to enhance the manuscript's quality as follows:

  1. The Author mentions on page 12, "dilution from 10to 10UFC/ml." Could you kindly clarify the precise meaning of "UFC" in this context?

R: Actually, is CFU, this was a mistake. The error was amended in the text.

  1. Please maintain a consistent format for "CFU/mL" or "CFU/ml" across the entire manuscript.

R: This section was amended in the manuscript as recommended by the reviewer.

  1. It is recommended that the Author includes a section discussing limitations and a clear conclusion in the manuscript to enhance comprehension.

 R: As recommended by the reviewer, the first paragraph of the discussion section is dedicated to the BIChromET plate limitations and the last paragraph is the study's conclusion.

  1. It is advisable to strictly follow the referencing and citation style specified by the journal's guidelines.

R: The citation style is going to be modified in the final version before publication by the journal editing service.

Reviewer 4 Report

Comments and Suggestions for Authors

I read with interest the paper by Ortiz de la Rosa et al. They evaluated the performance of a selective enriched medium for the detection of cefepime and piperacillin/tazobactam resistance/susceptibility in P. aeruginosa on bacterial strains and clinical respiratory samples collected at their centre.

The work is very interesting, diagnostic stewardship is an imperative topic nowadays and should also be of interest to general readers.

I have a few points to make to the authors that I hope will improve the quality of their work.

1) The absence of numbered lines hinders the suggestions I would like to submit to the authors, including English syntax and some mistakes. I hope the authors get a chance to review the text.

2) Introduction:

Figure 1. The prediction of being able to provide a result for “targeted therapy” after only 24 h (i.e. after parallel observation of results on BIChromET and conventional medium) is optimistic to say the least. By definition, targeted therapy can be implemented after the definitive AST results. In this case, the authors could refer to “optimal therapy”, i.e. the establishment of an antibiotic regimen based on non-final phenotypic results.

3) Results: The authors should review the abbreviations taking into account that the methods are at the end of the article. For example, see BMD in the results.

Both tables were named table 2, it is necessary to make explicit which results are from bacterial colony and which are directly from respiratory sample.

4) Discussion: “The use of the BIChromET plates as a complement to the traditional techniques may play an important role in clinical decision-making, optimizing antibiotic usage.”

This statement by the authors is very optimistic. How is it that the authors think they can optimise antibiotic therapy against P. aeruginosa using the results of this test? For example, in the case of co-resistance to cefepime and piperacillin/tazobactam, would they use carbapenems or ceftazidime? I think data from the local epidemiology of P. aeruginosa might help in this regard by providing a probabilistic estimate, but the authors made no reference to it.

Also, the limitations on strains with MICs close to the breakpoint could be very limiting: would the authors feel comfortable advising the clinician to keep, for example, piperacillin/tazobactam on a sensitive strain on BIChromET that would then prove to have a MIC of 16 mg/L?

The discussion on the limitations of the study and the costs of the test is very sketchy and I think needs to be improved.

I suggest the authors be less firm and decisive in their conclusions and revise the tenor of the text in accordance with the limitations of this diagnostic protocol.

Comments on the Quality of English Language

Extensive editing of English language required.

Author Response

I read with interest the paper by Ortiz de la Rosa et al. They evaluated the performance of a selective enriched medium for the detection of cefepime and piperacillin/tazobactam resistance/susceptibility in P. aeruginosa on bacterial strains and clinical respiratory samples collected at their centre.

The work is very interesting, diagnostic stewardship is an imperative topic nowadays and should also be of interest to general readers.

I have a few points to make to the authors that I hope will improve the quality of their work.

1) The absence of numbered lines hinders the suggestions I would like to submit to the authors, including English syntax and some mistakes. I hope the authors get a chance to review the text.

R: As the revision of English was recommended by the reviewer, our manuscript was checked by the English editing service.

2) Introduction:

Figure 1. The prediction of being able to provide a result for “targeted therapy” after only 24 h (i.e. after parallel observation of results on BIChromET and conventional medium) is optimistic to say the least. By definition, targeted therapy can be implemented after the definitive AST results. In this case, the authors could refer to “optimal therapy”, i.e. the establishment of an antibiotic regimen based on non-final phenotypic results.

R: This section was amended in the manuscript as recommended by the reviewer.

3) Results: The authors should review the abbreviations taking into account that the methods are at the end of the article. For example, see BMD in the results.

R: As suggested by the reviewer the manuscript was revised in order to avoid the appearance of abbreviations before the full words.

Both tables were named table 2, it is necessary to make explicit which results are from bacterial colony and which are directly from respiratory sample.

R: This was an error in the process of adaptation to the journal format. Sorry for the inconvenience. The mistake was amended in the manuscript. Additionally, we clarify in each table names which results were from bacterial colonies and which were from respiratory samples.

4) Discussion: “The use of the BIChromET plates as a complement to the traditional techniques may play an important role in clinical decision-making, optimizing antibiotic usage.”

This statement by the authors is very optimistic. How is it that the authors think they can optimise antibiotic therapy against P. aeruginosa using the results of this test? For example, in the case of co-resistance to cefepime and piperacillin/tazobactam, would they use carbapenems or ceftazidime? I think data from the local epidemiology of P. aeruginosa might help in this regard by providing a probabilistic estimate, but the authors made no reference to it.

R: Following the reviewer's recommendation, information about the local epidemiology and an explanation of how empirical treatment would be optimized has been added to the manuscript. Based on our epidemiological data (21%R meropenem and 22%R to ceftazidime versus 23%R piperacillin/tazobactam and 27%R cefepime) in case of co-resistance to TZP/FEP, meropenem would be the best choice although maybe is necessary to escalate to a wider spectrum antibiotic such as ceftazidime/avibactam or ceftolozane/tazobactam, until the antibiogram arrive.

Also, the limitations on strains with MICs close to the breakpoint could be very limiting: would the authors feel comfortable advising the clinician to keep, for example, piperacillin/tazobactam on a sensitive strain on BIChromET that would then prove to have a MIC of 16 mg/L?

R: Yes, since an isolate with MIC of 16 mg/L to TZP continues to be susceptible, high exposure (I, based on EUCAST). Following EUCAST recommendations, all P. aeruginosa isolates with MIC <= 16 mg/L should be treated with a high dose of antibiotic, so the dose to be used will not change. It should be noted that although it is not the perfect method, it is the best available so far with phenotypic results in 24 hours from direct sampling.

The discussion on the limitations of the study and the costs of the test is very sketchy and I think needs to be improved. I suggest the authors be less firm and decisive in their conclusions and revise the tenor of the text in accordance with the limitations of this diagnostic protocol.

R: As suggested by the reviewer, several limitations of the method were added to the discussion section, and the tenor of our conclusion was decreased.

Round 2

Reviewer 4 Report

Comments and Suggestions for Authors

The authors addressed properly all my comments.